# Assessing the low complexity of protein sequences via the low complexity triangle

**Pablo Mier** ⊙\*, **Miguel A. Andrade-Navarro**

Faculty of Biology, Institute of Organismic and Molecular Evolution, Johannes Gutenberg University Mainz, Mainz, Germany

\* munoz@uni-mainz.de

**Data Availability Statement:** All relevant data are within the manuscript. In addition, we have implemented a web server, LCT (http://cbdm-01. zdv.uni-mainz.de/~munoz/lct/), to host the code used in the analyses.

## Abstract

### Background

Proteins with low complexity regions (LCRs) have atypical sequence and structural features. Their amino acid composition varies from the expected, determined proteome-wise, and they do not follow the rules of structural folding that prevail in globular regions. One way to characterize these regions is by assessing the repeatability of a sequence, that is, calculating the local propensity of a region to be part of a repeat.

### Results

We combine two local measures of low complexity, repeatability (using the RES algorithm) and fraction of the most frequent amino acid, to evaluate different proteomes, datasets of protein regions with specific features, and individual cases of proteins with extreme compositions. We apply a representation called 'low complexity triangle' as a proof-of-concept to represent the low complexity measured values. Results show that proteomes have distinct signatures in the low complexity triangle, and that these signatures are associated to complexity features of the sequences. We developed a web tool called LCT (http://cbdm-01.zdv. uni-mainz.de/~munoz/lct/) to allow users to calculate the low complexity triangle of a given protein or region of interest.

### Conclusions

The low complexity triangle proves to be a suitable procedure to represent the general low complexity of a sequence or protein dataset. Homorepeats, direpeats, compositionally biased regions and globular regions occupy characteristic positions in the triangle. The described pipeline can be used to characterize LCRs and may help in quantifying the content of degenerated tandem repeats in proteins and proteomes.

**Funding:** This work was supported by Deutsche Forschungsgemeinschaft [AN735/4-1 to M.A.A.N.]. The funders had no role in study design, data collection and analysis, decision to publish, or preparation of the manuscript.

**Competing interests:** The authors have declared that no competing interests exist.

## Introduction

Amino acids are not part of proteins in equal proportions, as they do differ from the theoretical 5% usage that one would expect. The prevalence of the different amino acids in a proteome is taxonomically-dependent, yielding species-specific amino acid frequencies. However, about 20% of eukaryotic and 8% of prokaryotic residues are part of regions that do not follow the expected amino acid frequencies given by the full proteomes but contain fewer amino acid types [1]. These are known as low complexity regions (LCRs). They can be found in the form of consecutive runs of a single (homorepeat) or two ordered different residues (direpeats) [2], of multiple adjacent residue patterns (tandem repeats, [3]) and of scrambled arrangements of repetitive motifs (cryptic repeats, [4]). Because these regions are enriched in a few amino acid types they are also known as compositionally biased regions (CBRs) [5].

Low complexity regions are often functional [6,7] and do not follow the guidelines of structural folding that rule in globular regions [8]. This is because they usually overlap with protein disorder [9–12]. Sequence alterations in some of them have been associated with diseases [13,14]. There are many available computational tools to locate LCRs (e.g. [15–20]), but they are usually focused on one or a few types of low complexity classes. Efforts to integrate some of these tools into comprehensive low complexity tool platforms have also been made [21,22]. Similarly, there are dedicated databases of tandem repeats [23], disorder proteins [24,25], and other low complexity regions [26,27].

The low complexity diagram, or low complexity triangle, is a 2D representation useful to compare amino acid sequences of various degrees of complexity [5]. It represents the complexity composition of a sequence combining two local measurements: repeatability [28] and local fraction of the most frequent amino acid [5]. Similar to the use of the Ramachandran plot to visualize the values of backbone dihedral angles in protein structures [29], in the low complexity triangle the distribution of pairs of repeatability and amino acid frequency values found for a given sequence or dataset would be depicted, and a unique low complexity cloud or signature would emerge.

For a given sequence, to calculate its low complexity triangle one must first split it in overlapping windows, and then calculate two parameters for each window: the minimum number of mutations to perfect repeats (MMPR) and the fraction of the most frequent amino acid. The x-axis of the diagram shows the MMPR in the range from 0 to half of the window length. The y-axis represents the fraction of the most frequent amino acid in the current window, in the range from 1/window_length to 1. The prevalence distribution of the windows in the triangle would help in the assessment of the low complexity of the analyzed sequence, with regions of variable degrees of amino acid frequency (y-axis) and repetition (x-axis). Higher values in the y-axis force lower values in the x-axis, thus imposing the low complexity diagram to be drawn as a triangle [5].

Here we use the low complexity triangle as a proof-of-concept and study how it reflects low complexity features of protein sequences. We test it with several proteomes, datasets of protein regions with diverse complexity features, and some proteins with extreme compositions, and check that they have distinct signatures in the low complexity triangle depending on their profile. We developed the LCT tool to allow any user to submit their own protein of interest and calculate its associated triangle.

## Methods

### Data retrieval

The proteomes of *Escherichia coli* (TaxId = 83333), *Dictyostelium discoideum* (TaxId = 44689), *Saccharomyces cerevisiae* (TaxId = 559292), *Drosophila melanogaster* (TaxId = 7227) and

*Homo sapiens* (TaxId = 9606) were downloaded from UniProtKB release 2019_11, limiting the datasets to one protein sequence per gene. A total of 4391, 12739, 6049, 13790 and 20605 proteins were downloaded, respectively.

The complete SwissProt release 2019_11 (561568 proteins) was also retrieved from the UniProtKB database. Positional annotations from all proteins of that release were parsed out and several datasets covering different sequence features were generated; considered features were: domains, transmembrane regions, coiled coil regions and compositional bias. Per feature, only the amino acid positions of the specific annotated region were kept. Sequence features shorter than 20 amino acids were not considered, as this is the default window length for the repeatability analysis. As a result, we obtained 200406 domains (**S1 File**), 372581 transmembrane regions (**S2 File**), 22110 coiled coil regions (**S3 File**) and 59176 regions annotated as compositionally biased (**S4 File**).

To obtain a dataset of globular sequences, we downloaded all protein sequences from the PDB database [30], and randomly selected 1000 sequences to carry out the analysis (**S5 File**). A dataset of 100 manually-curated proteins containing low complexity regions was obtained from [5] (**S6 File**). From the database DisProt release 2020_06 [25], we retrieved a dataset of 6656 manually-curated disordered regions (**S7 File**).

### Low complexity assessment

Given a protein sequence, we first divided it in overlapping windows of 20 amino acids (default), with a step of one amino acid. Windows shorter than 20 amino acids (at the N- and C-terminal of the protein sequences) were discarded. Each window was then analyzed with the RES tool [28], which calculates the minimum number of mutations to perfect repeats (MMPR), $x$. RES is a simple and linearly complex tool useful to detect protein sequences approximate to short tandem repeats. It assesses the repeatability of an input sequence by quantifying how far it is from a perfect repeat. We used as input for RES all protein windows, one at a time, and received the resulting MMPR.

We also computed the fraction of the most frequent amino acid in the window, $y$. Once the complete set of windows was analyzed, we plotted the x/y values in a low complexity triangle. When dealing with a dataset of more than one protein, we divide each of them in windows and then analyze all windows. This is done to calculate the total amount of windows in the dataset that occupies each coordinate in the low complexity triangle, irrespective of protein length.

To ease the comparison between the distribution of the windows' results in the different low complexity triangles, we define three overlapping regions outlined with different colors: ($x > 8.5$, $y < 0.275$), in red; ($x > 7.5$, $y < 0.325$), in green; and ($x > 6.5$, $y < 0.375$), in orange.

### LCT server

The low complexity triangle tool can be executed using the LCT server (http://cbdm-01.zdv. uni-mainz.de/~munoz/lct/). As mandatory input, LCT needs a protein sequence in FASTA format, either pasted or uploaded (**Fig 1A**). The default window length (WL) is 20 amino acids, but the user can modify it to 10, 15, 25 or 30 amino acids. Smaller values of the window length would result in a shortened triangle's grid (i.e. for WL = 10, MMPR = [0–5]), and thus the resolution of the results would be poorer, but would allow the unequivocal identification of shorter fragments with repeats (top-left corner in the triangle). On the other hand, larger values of the window length would be useful to study the general complexity of a sequence and to uncover long tandem repeats, which would be placed closer to the bottom-left corner in the low complexity triangle.

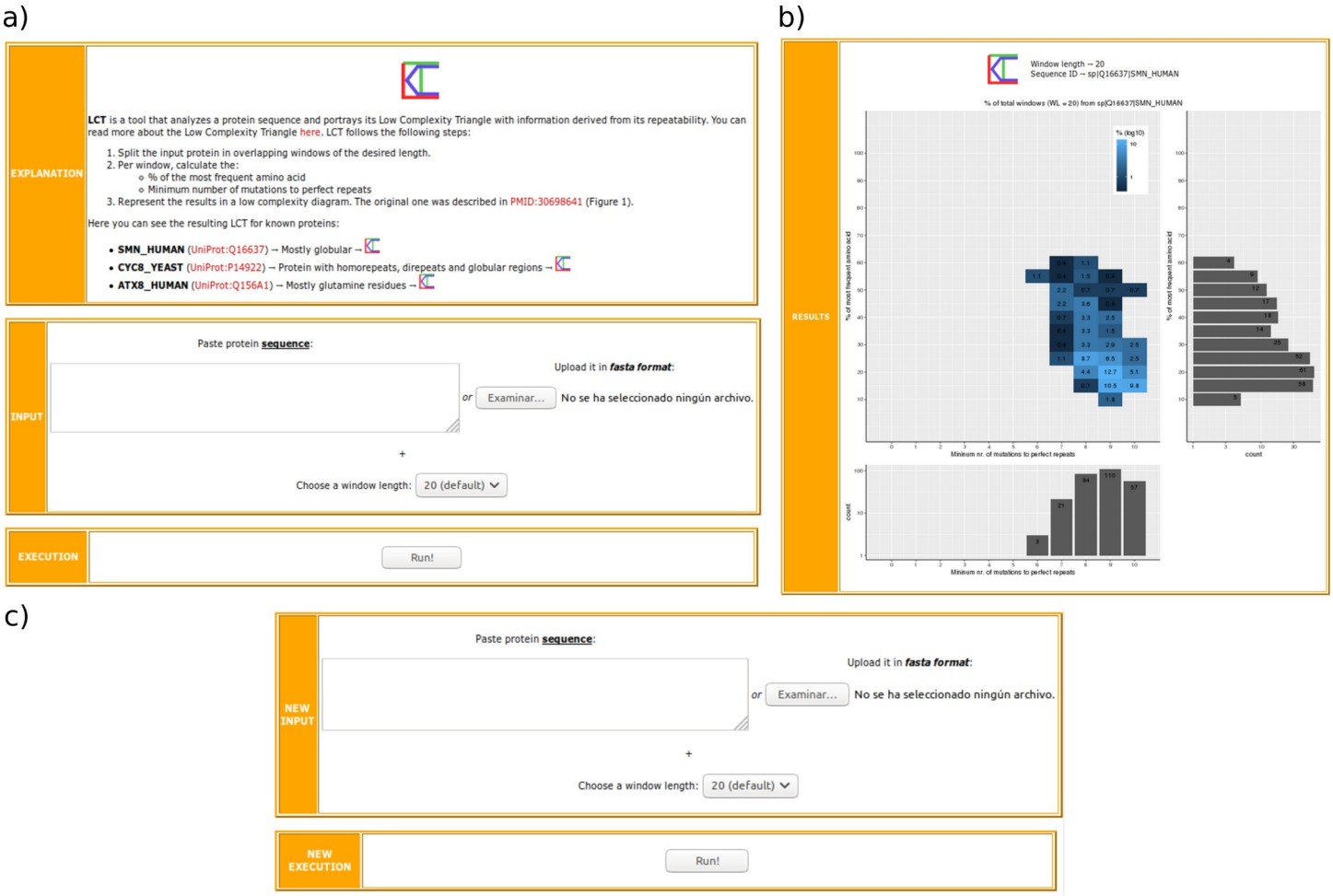

**Fig 1. LCT web tool.** a) Home page, b) results page, and c) iteration section in the LCT web tool.

There are some restrictions for the input sequences: 1) at least one sequence must be submitted to start the execution, 2) the query sequence must be a valid amino acid sequence, and 3) the query sequence needs to be at least 5 amino acids longer than the selected window length. When more than one sequence is used as input, restrictions apply for all proteins individually.

Results are shown in a separate window (**Fig 1B**). The low complexity triangle generated for the input sequence is displayed along with the chosen window length and the input sequence ID. LCT allows the user to easily reiterate its execution by letting the upload of a new input sequence, or possibly the same sequence with a different window length (**Fig 1A**). A standalone version of the code is available to download to study a larger dataset or proteome.

## Results & discussion

### Proteomes have distinct signatures in the low complexity triangle

The low complexity triangle is a plot meant to characterize and represent the low complexity content of a given sequence, region or dataset. Once the query is decomposed in overlapping windows, two distinct parameters related to its low complexity are calculated per window: repeatability and fraction of most frequent amino acid (see Methods). The distribution of the

pairs of values measured is displayed in a cartesian heatmap. Here we use as dataset the complete proteomes from several model organisms.

We used the 4391 proteins of the bacteria *Escherichia coli* as input and calculated the low complexity triangle for its complete proteome (**Fig 2A**). Their signal cloud (region where most of the windows are placed) is located at the bottom-right corner of the triangle. It means that, per window, a high number of residues would have to be mutated to obtain a perfect repeat

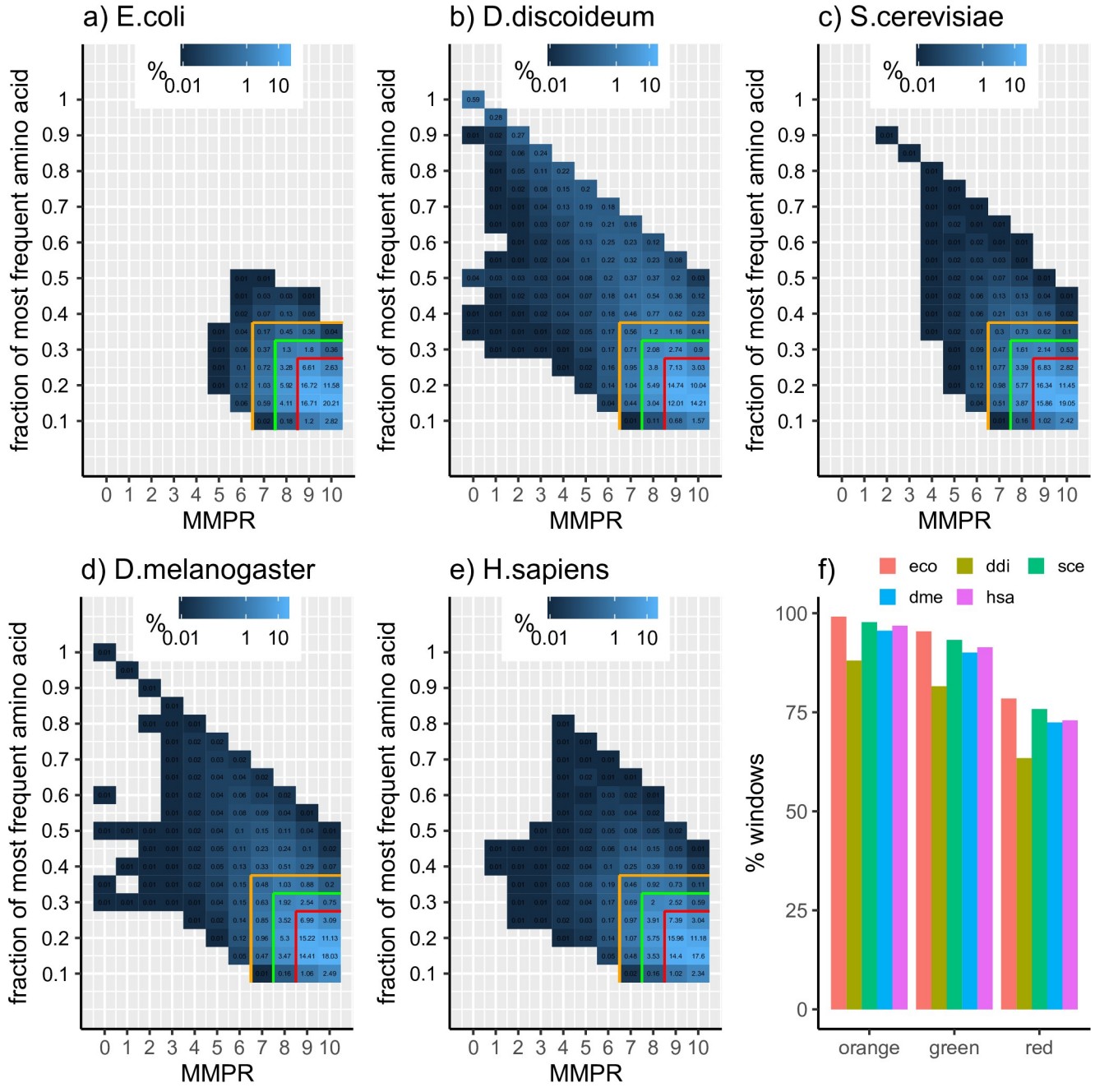

**Fig 2. Low complexity triangle for several proteomes.** a) *Escherichia coli*, b) *Dictyostelium discoideum*, c) *Saccharomyces cerevisiae*, d) *Drosophila melanogaster*, e) *Homo sapiens*, f) percentage of windows covered in the red, green and orange regions per species. MMPR = Minimum number of Mutations to Perfect Repeats. Eco = *E. coli*, ddi = *D. discoideum*, sce = *S. cerevisiae*, dme = *D. melanogaster*, hsa = *H. sapiens*. Window abundance shown in log10 scale. Window length = 20 amino acids.

(high x-axis values), and that amino acid proportions are close to random (low y-axis values). This result is in line with the reported low compositional bias of bacterial genomes [31,32].

The analysis of the amoeba *Dictyostelium discoideum* provides very different results. Its proteome is rich in repeats [33], showing a window frequency spread irradiating from the general unbiased bottom-right corner. Low x-axis values and/or high y-axis values represent low complexity regions of some sort. For example, the signal at y = 0.5, although weak, means that a part of the proteome is composed of direpeats, whereas the stronger y = 1 signal corresponds to homorepeat regions, also frequent in this proteome [32].

Three distinct species, *Saccharomyces cerevisiae*, *Drosophila melanogaster* and *Homo sapiens* have surprisingly similar low complexity triangles (**Fig 2C–2E**). However, the proportion of repetitive regions (x < = 3) is slightly higher in the proteome of the fruit fly and practically absent in the yeast proteome.

The numerical comparison of the previous low complexity triangles was done by studying how many windows cluster in the bottom-right corner of each plot. We calculated this value for three overlapping coordinate regions (orange, green and red; see Methods), obtained from **Fig 2A–2E**. *D. discoideum* has the lower amount of windows in these three regions (**Fig 2F**), and 12% of them are distributed in the plot outside the orange region, meaning more compositional bias and less globular regions. On the other hand, only 0.8% of the windows of *E. coli* are outside the orange region, indicating very little low complexity content. Results for *S. cerevisiae*, *D. melanogaster* and *H. sapiens* are relatively similar, having 2.3%, 4.4%, and 3.2% of their windows outside the orange region, respectively.

## Regions in the low complexity triangle are associated to sequence features

In general, as we have seen in the examples above, most proteins in proteomes are composed of globular regions. Their signature as a whole in the low complexity triangle is therefore mostly biased towards the bottom-right corner, where the globular regions are located. To show the power of the low complexity triangle to represent more diverse signatures of low complexity, we repeated the analysis done above with five different datasets: the complete SwissProt version 2019_11 and four datasets with protein regions corresponding to specific features annotated in SwissProt (see Methods).

The SwissProt database contains mainly globular proteins, therefore it could be taken as a proxy for a meta-proteome. We analyzed a complete SwissProt protein dataset and confirmed that their cloud signature (**Fig 3A**) is mostly similar to that of just protein domains (**Fig 3B**), plus weak signals for both direpeats (y = 0.5) and homorepeats (y = 1). Results for domain regions are exclusively limited to the bottom-right corner of the triangle, as expected, and are indeed confined to the orange region (99.3% of windows).

Transmembrane regions are hydrophobic sequences that span across the cell membrane. Although they are structured, their amino acid composition is largely biased towards hydrophobic residues. Their low complexity triangle resembles the one from globular domains, as expected, but there is one major difference: many windows are clustered in the orange region but not in the green one, and similarly in the green region but outside the red region (**Fig 3C**). Windows are distributed as 95.17%, 83.31% and 52.96% for transmembrane regions in the orange, green and red regions, respectively, comparatively lower than the 99.30%, 96.13% and 80.16% for domain regions. The reason for this contrast is that hydrophobic residues are more frequent in transmembrane regions [34], thus making these regions more compositionally biased, albeit not repeated, than other domains.

A similar case is that of coiled coil regions. They are heptad repeats of hydrophobic (h) and charged (c) amino acid residues following the pattern *hxxhcxc* [35]. Their overall values in the

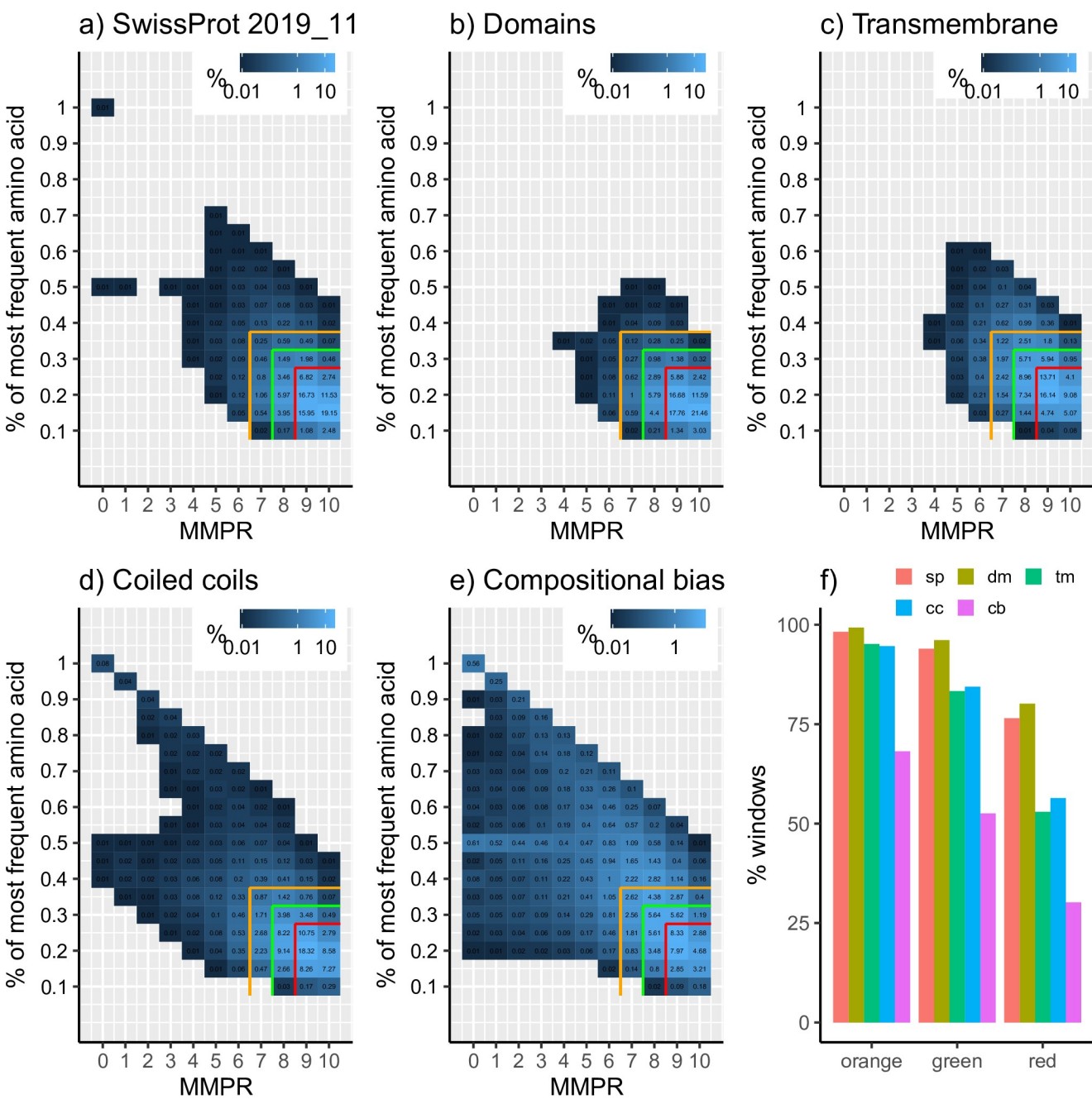

**Fig 3. Low complexity triangle for several protein datasets.** a) SwissProt release 2019_11, b) protein domains, c) transmembrane regions, d) coiled coil regions, e) compositionally biased regions, f) percentage of windows covered in the red, green and orange regions per dataset. MMPR = Minimum number of Mutations to Perfect Repeats. sp = SwissProt, dm = domains, tm = transmembrane, cc = coiled coils, cb = compositional bias. Window abundance shown in log10 scale. Window length = 20 amino acids.

low complexity triangle are highly comparable to the ones from transmembrane regions, although some of them have repeats (x < = 3) (**Fig 3D**). This does not necessarily have to be the case, as the repeated unit is the pattern of amino acid properties and not the amino acids themselves. The repeat signal is probably due to regions with repeats wrongly annotated as being coiled coils, even though we are limiting our study to SwissProt entries to minimize such

annotation errors. One example is the annotated coiled coil region in protein DDB_G0281095 from *D. discoideum* (UniProt:Q54UG3, positions 329–521). This region is highly glutamine-rich (146 glutamine residues in the 193-amino acid region) and contains long glutamine homorepeats (up to 58 consecutive glutamines). Glutamine homorepeats can adopt helical structure but this is position and context dependent [36].

In the last place we examined regions annotated as compositionally biased. In this case, there are windows occupying almost all of the possible places in the low complexity triangle (Fig 3E), with a clear shift in the distribution of the windows to the outside of the three colored regions. Only 30.19% of the windows analyzed in this dataset are in the red region (Fig 3F), and more than 31% of them populate coordinates outside the orange region. Signals for both direpeats ($y = 0.5$) and perfect homorepeats ($y = 1$) can also be distinguished.

We complemented this study analyzing with the low complexity triangle three manually-curated datasets from globular proteins, proteins with low complexity regions, and intrinsically disordered regions (S1 Fig). Values within the orange, green and red regions are highly similar between the set of globular proteins (98.19%, 94.52% and 78.27, respectively) and the domains obtained from SwissProt (99.30%, 96.13% and 80.16%; Fig 1A). The dataset of low complexity proteins yields the most sparse triangle, with 83.32%, 73.64% and 53.99% windows in the orange, green and red region, respectively. Even though disorder usually overlaps with low complexity, this is not always the case. As a result, windows analyzed from disorder regions are more scattered in the low complexity triangle than those from globular proteins, but less than pure low complexity regions.

## Applying the low complexity triangle to extreme protein regions

The low complexity triangle may also be applied to individual proteins or protein regions in them. Here we focus on three different examples covering the broad spectrum of signals the low complexity triangle can yield. We use specific regions from these proteins to show the signal clouds placed in the triangle associated directly to them, without any noise from the rest of the protein.

First, we focus on the simplest case, the expected outcome when analyzing the low complexity of a protein domain. Human SMN1 (UniProt:Q16637; SMN_HUMAN) is a nuclear protein involved in the splicing of cellular pre-mRNAs with a central Tudor domain. This domain (at amino acid positions 91–151) is structurally characterized (PDB:1MHN) and is a strongly bent anti-parallel beta-sheet consisting of five beta-strands with a barrel-like fold [37]. It is a globular domain that does not contain any repetitive element in its protein sequence, and thus the most frequent amino acid per window of length 20 is close to random ($y < = 0.2$) and the minimum number of mutations to perfect repeats is close to the maximum ($x > = 9$) (Fig 4A). The bottom right corner of the low complexity triangle is populated as expected, an extreme result similar to the one obtained for the complete set of domains (Fig 4B).

The alpha-1(VII) chain of the human collagen (UniProt:Q02388; CO7A1_HUMAN) is a structural constituent of the extracellular matrix. It is a long protein (2928 amino acids) with nine Fibronectin type III domains followed by a long triple helical region (positions 1254–2784). The latter is composed of a tripeptide repeating unit Gly-X-Y. When placed in the low complexity triangle, we see that results in the y-axis are mostly in the range 0.3–0.4 (Fig 4B). This is given by the glycine in the first position of the repeating unit. As the second and the third positions are variable, the values in the x-axis depend on them. As a result, the signal cloud for this query region is flat but spread. Results for tandem repeats will accumulate in the central part of the triangle.

Last we analyzed human ataxin-1 (UniProt:P54253; ATX1_HUMAN). It is a chromatin-binding factor mainly known and studied for being associated with human neurodegenerative

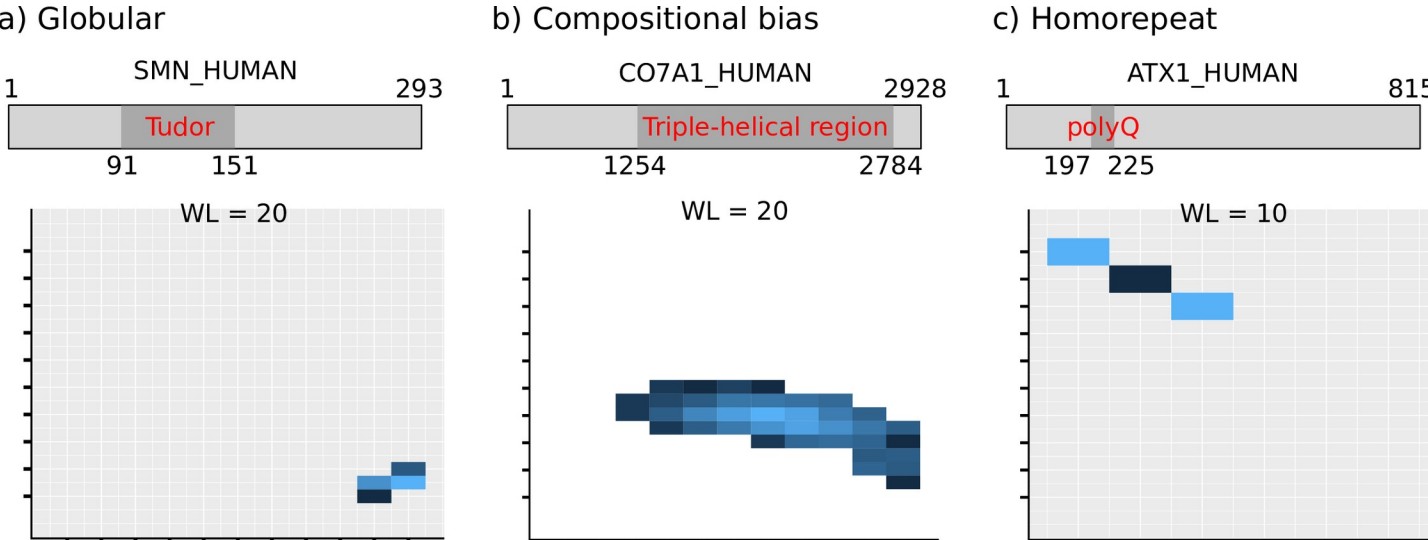

**Fig 4. Examples of extreme low complexity triangles.** a) Tudor domain from protein SMN_HUMAN (positions 91–151); globular. b) Triple-helical region from protein CO7A1_HUMAN (positions 1254–2784); compositionally biased. c) polyQ region from protein ATX1_HUMAN (positions 197–225); homorepeat; window length in this example is 10 amino acids, due to the shorter length of the selected region. Window abundance shown in log10 scale. WL = window length.

disease spinocerebellar ataxia 1 (SCA1) [38]. SCA1 is caused by the abnormal expansion of a polyglutamine (polyQ) region in positions 197–225, which leads to aggregation-induced neuronal toxicity. The wild-type polyQ contains 27 glutamines and two inserted histidine residues, $[Q]_{12}HQH[Q]_{14}$. To analyze this region with the low complexity triangle, we used window length = 10, as it is a relatively short region. The signal cloud is limited to the top-left corner of the triangle (**Fig 4C**). Values in the x-axis vary from 0, when the window is all glutamines, to 2, when the histidine residues are in the window, both of which should be mutated to glutamines to achieve a perfect repeat.

The repeatability of the previous examples is trivial. Globular domains do not usually contain repetitive regions, amino acid rich regions are compositionally biased, and homorepeats are repeats by definition. However, in many low complexity regions this is not the case. Either when facing a protein of unknown structure or to uncover degenerated tandem repeats, the low complexity assessment of a protein or protein region can be easily performed using the low complexity triangle.

## Using the LCT server

We have implemented a web server, LCT (http://cbdm-01.zdv.uni-mainz.de/~munoz/lct/), to host the code used in the analyses carried out in the previous section. We illustrate how the LCT server could be used to characterize low complexity in a query protein sequence. The human SARS coronavirus (SARS-CoV) (Severe Acute Respiratory Syndrome Coronavirus) nucleocapsid protein N (UniProt:P59595; NCAP_CVHSA) is a 422 amino acid long protein with two regions of known structure flanking a region annotated as serine-rich (**Fig 5A**). The LCT of the complete sequence suggests the existence of low complexity due to bias but without repeatability (**Fig 5B**). Separate analyses of the protein in three regions consisting of the N-terminal domain, the middle part, and the dimerizing C-terminal domain reveal that the low complexity resides mostly in the middle region (**Fig 5C**). This region is serine- and arginine-rich and was found to be responsible of the interaction of protein N with replicase subunit, nonstructural protein 3 (nsp3) in the related coronavirus mouse hepatitis virus (MHV) [39].

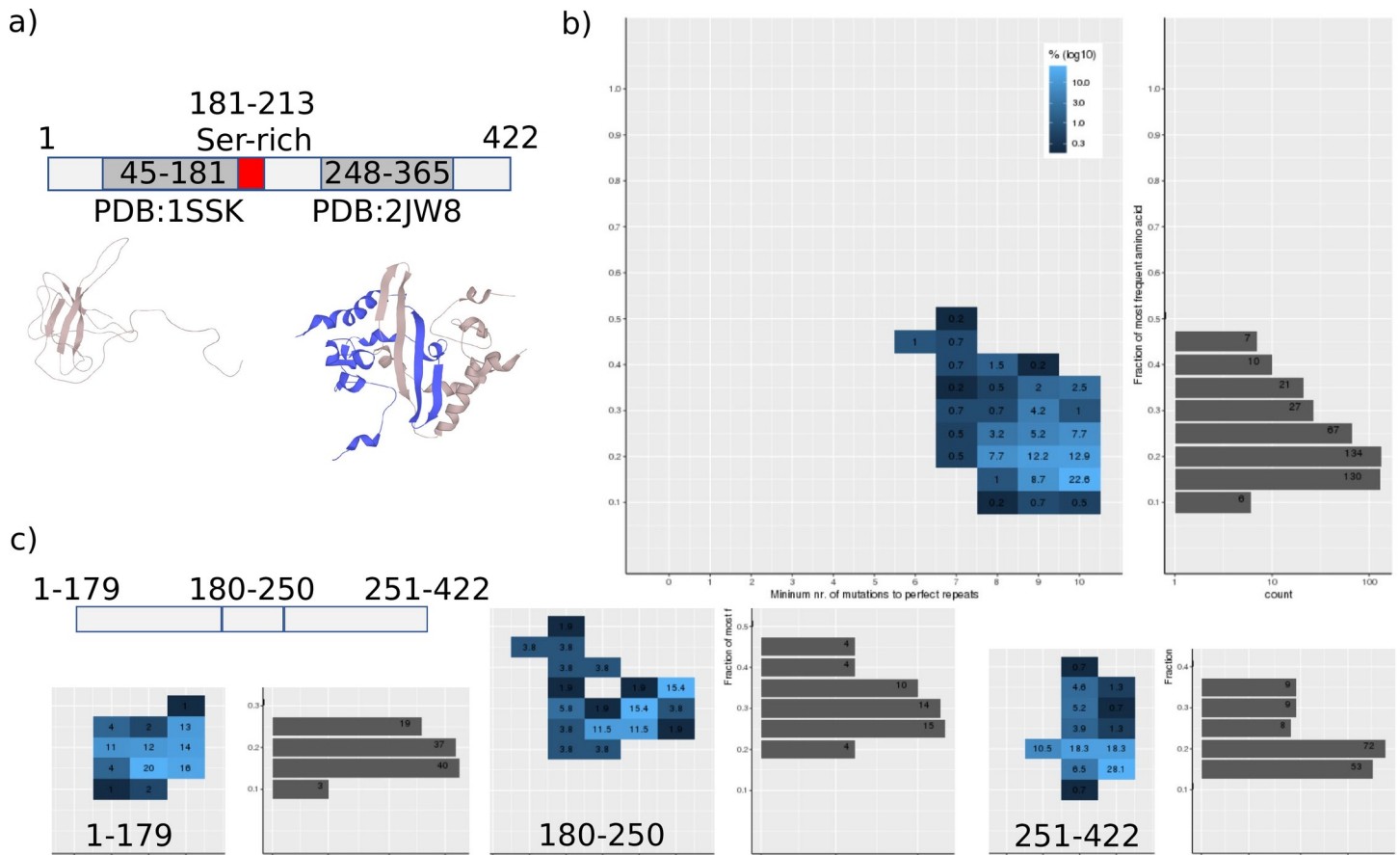

**Fig 5. LCT analysis of the nucleocapsid protein N from SARS-CoV.** a) The 422 amino acids long protein NCAP_CVHSA has an N-terminal RNA binding domain and a C-terminal homodimerizing domain, surrounding a region of uncharacterized structure that includes an annotated serine-rich region. b) LCT analysis of the complete sequence indicates the presence of low complexity without repeatability. c) Splitting the protein in three regions reveals that the low complexity content is placed in the middle structurally uncharacterized region.

The results obtained with the LCT server for this protein are comparable to those obtained when executing PlaToLoCo [22] with default parameters (S2 Fig). It calculates the low complexity regions from the input protein using several tools (SEG, CAST, fLPS, SIMPLE and GBSC). Most of them locate a region of low complexity in the central part of the NCAP_CVHSA protein, while others (GBSC and SEG-strict) do not detect any of these regions, and fLPS is highly unspecific.

## Conclusions

The low complexity triangle was described to visualize the types of low complexity present in protein sequences [5]. It takes into account two local sequence properties related to low complexity: The periodicity of the sequence and the fraction of the most frequent amino acid. The former is computed using a method called RES [28] to calculate sequence periodicity in terms of minimum number of mutations to perfect repeats (MMPR).

This visualization can be applied with, the fast and easy-to-use LCT server, which allows to analyze any given protein of interest with a customizable window length. A standalone version of the code is available to download to study a larger dataset or proteome (S8 File).

The low complexity triangle provides a quick and simple visual overview of the amount of low complexity in a protein, while indicating if this is including sequences with small repeats,

which are often found in such regions. The presence of such sequences can aid in the interpretation of the structural and functional properties of proteins. In addition to the application of the low complexity triangle to whole proteins, this can be easily applied to regions in proteins as a means to locate various types of low complexity regions. Moreover, the application of the low complexity triangle to sets of proteins, allows identifying properties of low complexity regions specific to the set. Here we have shown that if a proteome is analyzed, a species' usage of low complexity can be characterized; we suggest that it could as well be used to analyze other sets of proteins, for example those coded by differentially expressed genes in the study of an experiment following gene differential expression, or from a proteomics study.

We believe this is a step in the right direction for a quick and straightforward analysis that has the potential to uncover and characterize low complexity content in proteomes and proteins, thus aiding in the characterization of protein function and structure.

## Supporting information

**S1 Fig. Low complexity triangle for several datasets.** a) 1000 randomly selected globular proteins, from the PDB database, b) 100 manually-curated low complexity proteins, c) disordered regions from DisProt release 2020_06, d) percentage of windows covered in the red, green and orange regions per dataset. MMPR = Minimum number of Mutations to Perfect Repeats. Window abundance shown in log10 scale. Window length = 20 amino acids.
(PDF)

**S2 Fig. Results obtained from PlaToLoCo using protein NCAP_CVHSA (UniProt:P59595) as input, and default parameters.**
(PNG)

**S1 File. Set of domains obtained from SwissProt 2019_11.**
(ZIP)

**S2 File. Set of transmembrane regions obtained from SwissProt 2019_11.**
(ZIP)

**S3 File. Set of coiled coil regions obtained from SwissProt 2019_11.**
(ZIP)

**S4 File. Set of compositionally biased regions obtained from SwissProt 2019_11.**
(ZIP)

**S5 File. Set of 1000 randomly selected sequences from PDB.**
(ZIP)

**S6 File. Set of 100 manually-curated proteins containing low complexity regions.**
(ZIP)

**S7 File. Set of disordered regions from DisProt 2020_06.**
(ZIP)

**S8 File. Standalone version of the LCT tool.**
(ZIP)

## Author Contributions

**Conceptualization:** Pablo Mier, Miguel A. Andrade-Navarro.

**Data curation:** Pablo Mier.

**Formal analysis:** Pablo Mier.

**Funding acquisition:** Miguel A. Andrade-Navarro.

**Investigation:** Pablo Mier, Miguel A. Andrade-Navarro.

**Methodology:** Pablo Mier.

**Project administration:** Miguel A. Andrade-Navarro.

**Resources:** Pablo Mier.

**Software:** Pablo Mier.

**Supervision:** Miguel A. Andrade-Navarro.

**Validation:** Pablo Mier.

**Visualization:** Pablo Mier.

**Writing – original draft:** Pablo Mier.

**Writing – review & editing:** Pablo Mier, Miguel A. Andrade-Navarro.

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
