## [Decision Letter · Decision Letter 0]

5 Jun 2020

PONE-D-20-11896

Assessing the low complexity of protein sequences via the low complexity triangle

PLOS ONE

Dear Dr. Mier,

Thank you for submitting your manuscript to PLOS ONE. After careful consideration, we feel that it has merit but does not fully meet PLOS ONE’s publication criteria as it currently stands. Therefore, we invite you to submit a revised version of the manuscript that addresses the points raised during the review process.

We look forward to receiving your revised manuscript.

Kind regards,

Alexandre G. de Brevern, Ph.D.

Academic Editor

PLOS ONE

Journal Requirements:

'This work benefited from the Marie Skłodowska-Curie Research and Innovation Staff Exchange project “Repeat protein Function Refinement, Annotation and Classification of Topologies” (REFRACT), which received funding from the European Union’s Horizon 2020 research and innovation programme under the Marie Sklodowska-Curie grant agreement No. 823886.'

'This work was supported by Deutsche Forschungsgemeinschaft [AN735/4-1 to M.A.A.N.].

The funders had no role in study design, data collection and analysis, decision to publish, or preparation of the manuscript.'

Reviewers' comments:

Reviewer's Responses to Questions

**Comments to the Author**

1. Is the manuscript technically sound, and do the data support the conclusions?

Reviewer #1: Yes

Reviewer #2: Partly

2. Has the statistical analysis been performed appropriately and rigorously? 

Reviewer #1: Yes

Reviewer #2: N/A

3. Have the authors made all data underlying the findings in their manuscript fully available?

Reviewer #1: Yes

Reviewer #2: No

4. Is the manuscript presented in an intelligible fashion and written in standard English?

Reviewer #1: Yes

Reviewer #2: Yes

5. Review Comments to the Author

Reviewer #1: This is an interesting and important study that adds significantly to the field and will have a noticeable impact. The manuscript is well-written and concise. It adds significantly to the field and will have a noticeable impact. The developed tool will be a useful addition to the repertoire of modern computational biology. However, there are several.

1) I was surprised by not finding any references to the work conducted by Dr. Kajava, who developed several databases of protein repeats (e.g., RepeatsDB: a database of tandem repeat protein structures; PRDB: Protein Repeat DataBase) and several computational tools for ab initio identification of the tandem repeats (e.g., T-REKS: Tandem REpeats in sequences with a K-meanS based algorithm; TAPO: A combined method for the identification of tandem repeats in protein structures; Tally: a scoring tool for boundary determination between repetitive and non-repetitive protein sequences; and Tally-2.0: upgraded validator of tandem repeat detection in protein sequences).

2) A brief discussion should be added of the fact that the presence of protein repeats is often correlated with the presence of intrinsic disorder.

3) The authors should clearly define the meaning of the “Low complexity triangle”. A succinct explanation should be provided of this type of diagram and why it is called a triangle.

Reviewer #2: The manuscript details a new method for assessing the low complexity regions via the low complexity triangle.

It describes the method based on two metrics - repeatability and fraction of the most frequent amino acid in proteins

divided into sliding windows of specific length. Proposed method is applied on five different proteomes and proteins grouped according different features. Manuscript describes tool developed for presentation of low complexity triangle based on protein repeatability.

The manuscript does appear to have some scientific merit, although it is

hard to discern. It presents idea of using developed tool for assess which (part of) protein can be low complexity, but miss detail description of method. Also, the main part of manuscript (Results & discussion) in some parts seem a bit confusing. Comparisons of results obtained with proposed method and other related tools is missing.

The following are list of errors that need to be addressed before a full conclusion can be made:

1) Because the main part of assessing low complexity regions (LCR) is low complexity triangle and LCT tool, detail description of other available tools for location of LCR is necessary to include in manuscript: what are differences, capabilities, etc. This is cover by only one sentence in the manuscript. It will be better to make separate section and list some tools and their characteristics. This can be used in comparison of results obtained by proposed LCT server and some other tools - it is necessary to show that new method proposed by authors gives comparable or better results (than results obtained with already existing tools and methods).

2) Section 2. Methods include very few sentences about method used for assessing of protein low complexity. On the other side, LCT server description is put on section Results, which is inappropriate. LCT server can not be RESULT of "assessing the low complexity...." - topic mentioned int the manuscript title. If authors describe method, it is normally that description of server that implement/used for visualization of method results is put together.

3) In current version of manuscript larger part of section 2 is occupied with text related to data used in examples. It t is usually that section title will be "Material and methods" or similar.

Data retrieved from UniProtKB should be more precisely characterized. For example, there are many Escherichia coli genomes. Which one was used (K12 - which strain, O127H6, CFT073,....? It is necessary to specify organism identification. Also, it is necessary to include somewhere information about positional annotations of proteins related to the extracted groups (domains, transmembrane regions, ....). This is important if reader wants to repeat experiment and possibly compare results with results obtained from some other tool/method.

4) Method must be described in more detail. For example:

- Authors mentioned that divided windows are overlapped, but did not mentioned step in overlapping. Are windows just overlap without sliding or window of selected length slide across sequence?

- If input file includes more than one protein how calculation looks like? Does final results represent some kind average values of single proteins (related to overlapping or sliding windows) or all sequence are concatenated to big one which act as an input?

If user put the whole protein in one file (proteins placed consecutively one by one in fasta format) on which way such file is processed?

- Using different window length produce different results. What are differences among these results? Is there any benefit if default length is 20? And, why maximal length is restricted to 30?

5) In Method section, part 2.2 Low complexity assessment, last sentence in the paragraph:

"To ease the comparison between the distribution of the windows’ results in the different low complexity triangles, we draw three overlapping boxes in different colors in coordinates: (x > 8, y < 0.3), in red; (x > 7, y < 0.35), in green; and (x > 6, y <0.4), in orange."

There is not explanation why why exactly those boundaries are used. Also, on the figures orange, green and red lines are draw on different coordinates: red - (x>8.5, y>2.75), green (x>7.5, y>3.25) and orange (x>6.5, y>3.75)?!

6) About LCT server: authors offer possibility for download offline version, which is commendable. But, missing file with short description of package and list of prerequisites for installations (for example, on installation in Linux it is necessary to install biopython, ggplot, etc.)

On LCT server exists in-advance unknown/unpublished restriction. For example, if protein length is 21, default length is 20, server refused input with error message "The query sequence needs to be at least 5 amino acids longer than the selected window length. Try again." .

This message appear even if such protein exists in file together with bunch of other proteins (for example, for proteome taken as input)

All such restriction should be documented.

7) Section "Results & Discussion" is confusing in some parts. The principle of applying method on complete proteome if results are presented with average values is for discussion because each proteome consists of proteins with (very) different length and characteristics. What kind of normalization is taken into account? Does larger proteins influence in final results with higher weight? Or does the proteins with higher degree of low complexity "masks" proteins with lower level? Etc...

> "The numerical comparison of the previous low complexity triangles was done by studying how many windows cluster in the bottom-right corner of each plot."

It can not be seen precisely from the figures. LCT web server produce figures with numerical values (percent of windows) which does not exists on figures in the manuscript (why?). Missing numerical numbers prevent checking numbers presented in the manuscript.

From sentence "On the other hand, 99% of the windows of E. coli are within the orange region and therefore are mostly globular."

one can conclude that if large percent of widows fall into orange region than this protein (proteomes, ...) is globular. But from figure 2.d (coiled coils) visually it can be seen that most of windows also fall into orange region, but coiled coil proteins not preferred to be globular.

Also, material is taken from UniProtKb which contains mainly globular proteins(as mentioned in manuscript), so discussion that method discovered that they are mainly globular is strange.

As an example (in proof-of-concept) one protein or group of proteins) which are non-globular should be taken, and then compared results of globular/non-globular groups (orange, green, red lines, etc.). Also and more important it is necessary to take some proteins that can not be characterized as low complexity and analyze results and graphics. The manuscript proposed method for assessing low complexity proteins, so it is normal to show how method works on both low complexity proteins and opposite group.

8) Comparison of results obtained with some other tools that locate LCR should be included.

9) Conclusion should include concrete sentences about possible use of this method and site - who, why and what are the advantages of using this method for locating LCR

6. PLOS authors have the option to publish the peer review history of their article (what does this mean?). If published, this will include your full peer review and any attached files.

Reviewer #1: Yes: Vladimir N. Uversky

Reviewer #2: No

---

## [Author Response · Author response to Decision Letter 0]

3 Jul 2020

Authors’ response to reviews: 

We have responded to each of the points made by the reviewers, discussing both the issues raised and the changes made to address them. We listed responses and changes after each point.

We thank the referees for their insightful comments, which have certainly helped improve the manuscript.

###############

Reviewer #1:

This is an interesting and important study that adds significantly to the field and will have a noticeable impact. The manuscript is well-written and concise. It adds significantly to the field and will have a noticeable impact. The developed tool will be a useful addition to the repertoire of modern computational biology. However, there are several.

1) I was surprised by not finding any references to the work conducted by Dr. Kajava, who developed several databases of protein repeats (e.g., RepeatsDB: a database of tandem repeat protein structures; PRDB: Protein Repeat DataBase) and several computational tools for ab initio identification of the tandem repeats (e.g., T-REKS: Tandem REpeats in sequences with a K-meanS based algorithm; TAPO: A combined method for the identification of tandem repeats in protein structures; Tally: a scoring tool for boundary determination between repetitive and non-repetitive protein sequences; and Tally-2.0: upgraded validator of tandem repeat detection in protein sequences).

The reviewer is completely right, we missed some important references in the original manuscript. To cover the current status of the low complexity field in a more comprehensive way, we have included more references in the Introduction of the revised manuscript, plus information about dedicated databases of low complexity regions.

2) A brief discussion should be added of the fact that the presence of protein repeats is often correlated with the presence of intrinsic disorder.

We have included a comment on this in the Introduction of the revised manuscript.

3) The authors should clearly define the meaning of the “Low complexity triangle”. A succinct explanation should be provided of this type of diagram and why it is called a triangle.

We have included comments on both issues in the Introduction of the revised manuscript.

###############

Reviewer #2:

The manuscript details a new method for assessing the low complexity regions via the low complexity triangle. It describes the method based on two metrics - repeatability and fraction of the most frequent amino acid in proteins divided into sliding windows of specific length. Proposed method is applied on five different proteomes and proteins grouped according different features. Manuscript describes tool developed for presentation of low complexity triangle based on protein repeatabilty. The manuscript does appear to have some scientific merit, although it is hard to discern. It presents idea of using developed tool for assess which (part of) protein can be low complexity, but miss detail description of method. Also, the main part of manuscript (Results & discussion) in some parts seem a bit confusing. Comparisons of results obtained with proposed method and other related tools is missing. The following are list of errors that need to be addressed before a full conclusion can be made:

1) Because the main part of assessing low complexity regions (LCR) is low complexity triangle and LCT tool, detail description of other available tools for location of LCR is necessary to include in manuscript: what are differences, capabilities, etc. This is cover by only one sentence in the manuscript. It will be better to make separate section and list some tools and their characteristics. This can be used in comparison of results obtained by proposed LCT server and some other tools - it is necessary to show that new method proposed by authors gives comparable or better results (than results obtained with already existing tools and methods).

We have included an additional analysis in Results section “LCT server”. Now we also analyze the NCAP_CVHSA protein with the PlaToLoCo webserver, which allows us to compare the results from the LCT algorithm to those from many other tools for finding low complexity regions (SEG, CAST, fLPS, SIMPLE and GBSC).

2) Section 2. Methods include very few sentences about method used for assessing of protein low complexity. 

We have included further explanations on RES in the revised manuscript, Methods section “Low complexity assessment”.

*) On the other side, LCT server description is put on section Results, which is inappropriate. LCT server can not be RESULT of "assessing the low complexity...." - topic mentioned int the manuscript title. If authors describe method, it is normally that description of server that implement/used for visualization of method results is put together.

Following the suggestion of the reviewer, we have relocated the LCR server explanation to a new Methods section “LCR server”. 

3) In current version of manuscript larger part of section 2 is occupied with text related to data used in examples. It t is usually that section title will be "Material and methods" or similar.

We believe that this text, 1-2 sentences per dataset explaining its features, is important to understand the results obtained in the analysis and should remain in the Results section.

*) Data retrieved from UniProtKB should be more precisely characterized. For example, there are many Escherichia coli genomes. Which one was used (K12 - which strain, O127H6, CFT073,....? It is necessary to specify organism identification. Also, it is necessary to include somewhere information about positional annotations of proteins related to the extracted groups (domains, transmembrane regions, ....). This is important if reader wants to repeat experiment and possibly compare results with results obtained from some other tool/method.

To unequivocally identify the species used in the analysis, the taxonomic identifier of each of them was added to its description in Methods section “Data retrieval”.

We also included as Supplementary Files 1-4 the set of domains, transmembrane, coiled coil and compositionally biased regions parsed out from the SwissProt release 2019_11 used in the analysis. 

4) Method must be described in more detail. For example:

- Authors mentioned that divided windows are overlapped, but did not mentioned step in overlapping. Are windows just overlap without sliding or window of selected length slide across sequence?

- If input file includes more than one protein how calculation looks like? Does final results represent some kind average values of single proteins (related to overlapping or sliding windows) or all sequence are concatenated to big one which act as an input?

If user put the whole protein in one file (proteins placed consecutively one by one in fasta format) on which way such file is processed?

We have extended the Methods section “Low complexity assessment” to include further clarification on all aspects suggested by the reviewer.

- Using different window length produce different results. What are differences among these results? Is there any benefit if default length is 20? And, why maximal length is restricted to 30?

This point is already discussed in the manuscript, in Methods section “LCT server”. The text reads: “The default window length (WL) is 20 amino acids, but the user can modify it to 10, 15, 25 or 30 amino acids. Smaller values of the window length would result in a shortened triangle’s grid (i.e. for WL = 10, MMPR = [0-5]), and thus the resolution of the results would be poorer, but would allow the unequivocal identification of shorter fragments with repeats (top-left corner in the triangle). On the other hand, larger values of the window length would be useful to study the general complexity of a sequence and to uncover long tandem repeats, which would be placed closer to the bottom-left corner in the low complexity triangle.”

5) In Method section, part 2.2 Low complexity assessment, last sentence in the paragraph:

"To ease the comparison between the distribution of the windows’ results in the different low complexity triangles, we draw three overlapping boxes in different colors in coordinates: (x > 8, y < 0.3), in red; (x > 7, y < 0.35), in green; and (x > 6, y <0.4), in orange.". There is not explanation why why exactly those boundaries are used. Also, on the figures orange, green and red lines are draw on different coordinates: red - (x>8.5, y>2.75), green (x>7.5, y>3.25) and orange (x>6.5, y>3.75)?!

We have clarified this description accordingly in the Methods section of the manuscript as follows: “we define three overlapping regions outlined with different colors: (x > 8.5, y < 0.275), in red; (x > 7.5, y < 0.325), in green; and (x > 6.5, y < 0.375), in orange.” We also changed all uses of the word “box” to “region”.

6) About LCT server: authors offer possibility for download offline version, which is commendable. But, missing file with short description of package and list of prerequisites for installations (for example, on installation in Linux it is necessary to install biopython, ggplot, etc.).

On LCT server exists in-advance unknown/unpublished restriction. For example, if protein length is 21, default length is 20, server refused input with error message "The query sequence needs to be at least 5 amino acids longer than the selected window length. Try again.". This message appear even if such protein exists in file together with bunch of other proteins (for example, for proteome taken as input).

All such restriction should be documented.

We thank the reviewer for pointing out both issues. We have improved the already available README file within the downloadable package with information about: Description, Dependencies, Usage, Example, Output. Also, the reviewer is completely right in that input restrictions in the server were not properly documented in the manuscript. We have included them both in the revised manuscript (Methods section “LCT server”) and in the server.

7) Section "Results & Discussion" is confusing in some parts. The principle of applying method on complete proteome if results are presented with average values is for discussion because each proteome consists of proteins with (very) different length and characteristics. What kind of normalization is taken into account? Does larger proteins influence in final results with higher weight? Or does the proteins with higher degree of low complexity "masks" proteins with lower level? Etc…

Per protein, we divide it in overlapping windows of the given window length. This is done for all proteins in a proteome or dataset. Then, for each window (irrespective of its protein of origin) we calculate parameters x (minimum number of mutations to perfect repeats) and y (fraction of the most frequent amino acid in the window). With this procedure we can calculate the total amount of windows in the dataset that occupies each coordinate in the low complexity triangle. We have included a clarification on this matter in Methods section “Low complexity assessment”.

*) "The numerical comparison of the previous low complexity triangles was done by studying how many windows cluster in the bottom-right corner of each plot.". It can not be seen precisely from the figures. LCT web server produce figures with numerical values (percent of windows) which does not exists on figures in the manuscript (why?). Missing numerical numbers prevent checking numbers presented in the manuscript.

Originally, we left the numbers out of the figures because they are too small to be read; we thought the color code would be enough. The revised Figures 2 and 3 now contain these numbers. 

*) From sentence "On the other hand, 99% of the windows of E. coli are within the orange region and therefore are mostly globular.", one can conclude that if large percent of widows fall into orange region than this protein (proteomes, ...) is globular. But from figure 2.d (coiled coils) visually it can be seen that most of windows also fall into orange region, but coiled coil proteins not preferred to be globular.

We have modified the sentence that now reads “On the other hand, only 0.8% of the windows of E. coli are outside the orange region, indicating very little low complexity content”.

*) Also, material is taken from UniProtKb which contains mainly globular proteins (as mentioned in manuscript), so discussion that method discovered that they are mainly globular is strange.

In Results section “Regions in the low complexity...” we use SwissProt for all analysis; testing SwissProt as a whole as a first step acts as a dataset check. In the manuscript we state this as “We analyzed a complete SwissProt protein dataset and confirmed that their cloud signature (Fig 3a) is mostly similar to that of just protein domains...”. It is not a discussion rather than a confirmation, before parsing out datasets for the different sequence features. Then with the different panels in Figure 3 we show that SwissProt does not only contain globular regions.

*) As an example (in proof-of-concept) one protein or group of proteins) which are non-globular should be taken, and then compared results of globular/non-globular groups (orange, green, red lines, etc.). Also and more important it is necessary to take some proteins that can not be characterized as low complexity and analyze results and graphics. The manuscript proposed method for assessing low complexity proteins, so it is normal to show how method works on both low complexity proteins and opposite group.

Following the recommendation of the reviewer, we have included an additional analysis of three manually-curated protein datasets: globular proteins, low complexity proteins, and disordered regions. Results are added in section “Regions in the low complexity triangle are associated to sequence features”, and new supplementary files (S5-S7_File) and figure (S1_Figure) have been included in the revised manuscript.

8) Comparison of results obtained with some other tools that locate LCR should be included.

See reply to comment #1.

9) Conclusion should include concrete sentences about possible use of this method and site - who, why and what are the advantages of using this method for locating LCR

We have expanded the section Conclusions to address the request of the reviewer.

---

## [Decision Letter · Decision Letter 1]

1 Sep 2020

Assessing the low complexity of protein sequences via the low complexity triangle

PONE-D-20-11896R1

Dear Dr. Mier,

We’re pleased to inform you that your manuscript has been judged scientifically suitable for publication and will be formally accepted for publication once it meets all outstanding technical requirements.

Kind regards,

Alexandre G. de Brevern, Ph.D.

Academic Editor

PLOS ONE

Additional Editor Comments (optional):

Reviewers' comments:

Reviewer's Responses to Questions

**Comments to the Author**

1. If the authors have adequately addressed your comments raised in a previous round of review and you feel that this manuscript is now acceptable for publication, you may indicate that here to bypass the “Comments to the Author” section, enter your conflict of interest statement in the “Confidential to Editor” section, and submit your "Accept" recommendation.

Reviewer #1: All comments have been addressed

Reviewer #2: All comments have been addressed

2. Is the manuscript technically sound, and do the data support the conclusions?

Reviewer #1: Yes

Reviewer #2: Yes

3. Has the statistical analysis been performed appropriately and rigorously? 

Reviewer #1: Yes

Reviewer #2: N/A

4. Have the authors made all data underlying the findings in their manuscript fully available?

Reviewer #1: Yes

Reviewer #2: Yes

5. Is the manuscript presented in an intelligible fashion and written in standard English?

Reviewer #1: Yes

Reviewer #2: Yes

6. Review Comments to the Author

Reviewer #1: In my view, all issues pointed by the reviewers were adequately addressed and the manuscript was revised accordingly.

Reviewer #2: Authors respond to all reviewer's comments. Comment on sentence:

"Originally, we left the numbers out of the figures because they are too small to be read; we thought the color code would be enough. The revised Figures 2 and 3 now contain these numbers."

Suggestion is to implement possibility to save figures generated on server in some other format (now it is possible to save it in .png).

Although current format can also be zoomed to clear read numbers, it will be better (also for possible future users) to implement save option to allow user to save figure in some other format (for example jpg, eps, ..) which is more convenient for for various purposes.

7. PLOS authors have the option to publish the peer review history of their article (what does this mean?). If published, this will include your full peer review and any attached files.

Reviewer #1: **Yes: **Vladimir N. Uversky

Reviewer #2: No

---

## [Editor Report · Acceptance letter]

16 Sep 2020

PONE-D-20-11896R1 

Assessing the low complexity of protein sequences via the low complexity triangle 

Dear Dr. Mier:

I'm pleased to inform you that your manuscript has been deemed suitable for publication in PLOS ONE. Congratulations! Your manuscript is now with our production department. 

Kind regards, 

on behalf of

Dr. Alexandre G. de Brevern 

Academic Editor

PLOS ONE